# How Lazy Reading and Semantic Sloppiness May Harm Progress in Synucleinopathy Research

**DOI:** 10.3390/biom12020228

**Published:** 2022-01-28

**Authors:** Erwan Bezard

**Affiliations:** 1Institut des Maladies Neurodégénératives, UMR 5293, Université de Bordeaux, 33076 Bordeaux, France; erwan.bezard@u-bordeaux.fr; 2CNRS, Institut des Maladies Neurodégénératives, UMR 5293, 33076 Bordeaux, France

**Keywords:** Parkinson’s disease, α-synuclein, amyloid, aggregation

## Abstract

While confronted with the increasing complexity of the neurobiology of Parkinson’s disease (PD), we face the ever-increasing sloppiness of the conceptual definitions associated with poor methodological characterizations and the use of unacknowledged proxies, all of which are harmful contributors to the overall slow progress of PD research. In this opinion paper, I share part of my frustration, acknowledge how I participate in this trend, and propose a simple remedy. Fighting against semantic or conceptual sloppiness is of paramount importance, notably for the benefit of newcomers to the field who otherwise would take for granted the classic assertions found *ad nauseam* in the literature.

## 1. Introduction

After 25 years of research in the Parkinson’s disease (PD) field (and over 250 published papers), every day I find it more challenging to read PD-related scientific papers. On the one hand, we are confronted with the increasing complexity of the neurobiology of PD. On the other hand, we face the ever-increasing slacking off with the conceptual definitions associated with poor methodological characterizations and the use of unacknowledged proxies. All of these factors are harmful contributors to the overall slow progress of PD research. In this opinion paper, I share my frustration, acknowledge how I participate in this trend, and propose remedies. Fighting against semantic and conceptual sloppiness is of paramount importance, notably for the benefit of newcomers to the field who otherwise would take for granted the classic assertions found *ad nauseam* in the literature.

## 2. Contribution to the Debate

Frustration often starts with the very first sentences that introduce PD as, for instance, “characterized by symptoms such as tremor, rigidity, and slowness of movement caused by the decline in dopamine neurotransmission in the striatum, a result of loss of dopaminergic neurons in the substantia nigra pars compacta”, a statement often supported by ancient references. The frustration here is two-fold. First, more recent collective contributions indicate that PD begins with a focal onset of the cardinal motor features (akinesia, rigidity, and tremor) [1,2]. In addition, PD happens to be heterogeneous, with a combination of motor and non-motor features [3]. Berg et al. is, for instance, the latest consensual description with high-specificity criteria for de novo PD according to the International Parkinson and Movement Disorder Society [2].

Second is the implicit suggestion that PD symptomatology is restricted to the loss of dopaminergic neurons in the ventrolateral area of the substantia nigra pars compacta (SNc). The hallmarks of any neurodegenerative disorder (PD does not escape from this statement) are neuronal loss, axonal and synaptic decline, and the associated neuroglia reaction. Authors have no problem implicitly relating the presence of rest tremor (often cited first although present in only 40% of PD patients at onset) to nigrostriatal degeneration. However, tremor is poorly (if at all) sensitive to dopamine replacement therapy. At the same time, it is improved by deep brain stimulation of the ventral intermediate nucleus of the thalamus [4], a relay of cerebellum inputs to the cortices. Cell and/or terminal loss of the pedunculopontine nucleus, locus coeruleus nuclei, dorsal motor nucleus of the vagus nerve, raphe nuclei, nucleus basalis of Meynert, ventral tegmental area, thalamus, hypothalamus, olfactory bulb, and some cortices in the late stage, is likely to contribute to some of the other symptoms of PD and, most notably, to the myriad of non-motor features [5,6].

In addition to neuronal loss, a defining feature of PD is the appearance of proteinaceous, α-synuclein (α-syn) rich, lipidous and organellar inclusions, called Lewy pathology, under the form of Lewy bodies (LBs) into the cytoplasm and Lewy neurites (LNs) into the axon, exclusively in about 15% of the SNc remaining neurons (a proportion that is relatively stable throughout the disease duration) [7]. While a rigorous determination of the regional loss of neurons in post-mortem tissue has been difficult, the advent of immunohistochemical techniques, combined with classic histological stainings, allowing the for localization of “both normal and abnormal forms” of α-syn has propelled the study of Lewy pathology forward [8]. Two decades ago, Braak and colleagues used these approaches to compare brains taken from asymptomatic individuals and PD patients at various times after diagnosis, leading them to hypothesize that Lewy pathology spreads into the brain from either the olfactory bulb or the dorsal motor nucleus of the vagus in the caudal medulla, two brain regions with axons extending to body surface [8].

Over the past two decades, the highly cited Braak hypothesis has dominated the field of PD pathogenesis [8] and received some experimental support in rodent [9,10] and non-human primate [11,12,13] models. According to this hypothesis, α-syn pathology would first begin in the peripheral autonomic nervous system and dorsal motor nucleus of the vagus or the olfactory bulb, and then spread predictably and sequentially to the upper brainstem and the cerebral hemispheres [14]. This hypothesis is attractive because (1) the early non-motor manifestations may precede nigrostriatal degeneration and the clinical development of PD; (2) Braak staging might correlate with non-motor manifestations and explain the clinical evolution of PD; and (3) a peripheral origin with a bottom-up propagation of α-syn toxic species could be a mechanism contributing to the onset and progression of PD and pave the way into effective neuroprotective therapies. However, the original authors critically discussed their hypothesis, expressing caution against simplistic generalization [15]. The harm was, however, already done. Sadly, the theory, not even propelled by the original authors, is, chiefly, accepted plainly, leading to dogmatism about the origin and progression of PD. The dogmatic hypothesis essentially ignores that the Braak et al. staging is a map of α-syn immunoreactivity that does not match neuronal loss [5,6] which, by the way, was not even studied in the original contribution [8]. The dogmatic hypothesis takes for granted a series of events whereby α-syn toxicity leads to neuronal degeneration and the appearance of clinical manifestations. The truth is that a good part of these assumptions is open to question when applied to the PD or experimental PD brains.

Interestingly, the immunohistochemical approach chosen by Braak and colleagues was made possible thanks to the seminal, although super-short, publication of Spillantini et al., back in 1997 [16]; it was a work considered by many (including myself) as one of the most influential papers in the last 25 years in PD research [17]. The authors elegantly demonstrated that the LBs and LNs found in the brainstem and cortices of both PD and dementia with Lewy bodies cases were firmly immuno-positives for α-syn using various anti-α-syn antibodies (raised against peptides corresponding to residues 116–131 and residues 11–34 of human α-syn); a finding, when associated with the then-recent genetic association of α-syn mutation and familial parkinsonism [18], which suggested that both diseases might be considered as synucleinopathies. Aggregation was implicit as α-syn immunoreactivity was found in anatomopathological landmarks (for instance, LBs, LNs, and glial cytoplasmic inclusions) that were otherwise stained with amyloid dyes [8,16,19,20]. A classic antibody used in many anatomopathological studies is still LB509 that reacts with an epitope located in the region encoded by amino acids 115–122 of α-syn, i.e., very much like the 116–131 antibodies used by Spillantini [16] and Braak [8]. Note, however, that there were no mentions of aggregation, conformation specificity, or post-translational modification.

Introductions often state that α-syn is the main or a major constituent of LBs, glial cytoplasmic inclusions, or LNs. Although not wrong, the use of words, such as “main” or “major”, implicitly suggests that α-syn represents over 50% of the content of such intracellular bodies, leading junior readers to consider LBs, glial cytoplasmic inclusions, or LNs as primarily made of aggregated α-syn with some partners classically used alongside for characterizing them. However, LBs or glial cytoplasmic inclusions are complex intracellular inclusions. LBs, for instance, are eosinophilic inclusions comprising filamentous structures in a dense core surrounded by a peripheral halo [21] containing numerous membranous components and dysmorphic organelles [22], i.e., far from being a simple protein waste bag. The proteomic analysis of fractions enriched in LBs from PD brains or glial cytoplasmic inclusions from MSA brain by different methods identified hundreds of proteins [23,24,25,26], in addition to α-syn. In none of these approaches did α-syn represented 50% of the content. In the most recent study performing a quantitative comparison of the proteomes composing LBs and glial cytoplasmic inclusions, Laferriere et al. identified 1022 proteins resisting sarkosyl solubilization in these inclusions [27]. α-syn is by far the most significantly enriched protein in both types of samples, with a fold change of disease vs. control close to 40, while the second most enriched protein reaches a fold change of 10 [27]. Therefore, we could state that α-syn is a prominent/overly represented constituent, but also emphasize the complex proteome of these inclusions, an overlooked information of paramount importance for understanding the nature and origin of these intracellular inclusions. Further caution is required with the above assumption that LBs are mainly (if not exclusively) of a proteinaceous nature. The ground-breaking study by Shahmoradian et al. extends beyond that by pointing at lipids/organelles and questioning the fibrillar character of α-syn in the lesions [22]. If these colleagues are right (and a few others since this publication), then α-syn lesions are—at least to a large extent—not proteinaceous aggregates. While the jury is out (maybe it will never return), it seems like the proteinaceous character of LBs should be introduced with some caution.

Fujiwara and colleagues first evoked the possible importance of α-syn phosphorylation at serine 129 (pSer129) in 2002 [28]. They showed, by mass spectrometry analysis and immunocytochemistry with an antibody that explicitly recognizes pSer129 α-syn, that this residue is selectively and extensively phosphorylated in synucleinopathy lesions. Their anatomopathological demonstration in post-mortem tissues and two transgenic mouse lines was accompanied by an in vitro investigation of the impact of pSer129 upon α-syn fibrillization, primarily assessed through Thioflavin T fluorescence [28]. Allow me to cite the cautious conclusion of the authors: “*our observations strongly suggest that excessive phosphorylation of α-syn in synucleinopathy brains (including those in transgenic animals), as well as the underlying alterations in the activities of various kinases and phosphatases, and in the conformation of α-syn, may contribute to abnormalities of α-syn. In conjunction with other post-translational modifications of α-syn, these changes would cause aggregation, eventually leading to neuronal cell death* [28].” What has eventually been understood by the field from this original publication can be paraphrased as follows: “*α-syn phosphorylation is equivalent to α-syn aggregation, which leads to neuronal cell death*.” From 2002, in the overwhelming majority of post-mortem and experimental studies, the assessment of α-syn phosphorylation was taken as an indirect, but a supposedly reliable, marker of aggregation without actually stating the proxy nature of the endpoint, proxy character that was already far-fetched, as seen above. There is therefore much in the clinical pathology literature to draw on and be more precise about the cytopathological (cytological) definition of synucleinopathy. In fact, many papers have been satisfied with even very diffuse pSer129 positivity to qualify experimental histological images as synucleinopathy without any consideration of the presence of real subcellular inclusions in the measured/quantified signal. This notion allows stratification of the “sloppiness” in the analysis of experimental synucleinopathies, highlighting that the semantic and conceptual efforts should be made by both scientists working with human tissues and experimental/animal material.

The subsequent proxy was then to take the measurement of α-syn phosphorylation by any methodology using various, more or less validated, antibodies as a “sound” assessment of “synucleinopathy”. A dedicated paper would be needed for covering all the technical aspects and the myriad of antibodies reported in use in the literature. The move towards adopting such a sloppy proxy was oblivious of a debate for deciding whether phosphorylation at Ser129 suppresses or enhances α-syn aggregation and toxicity in vivo [29], a debate that is particularly intense when considering the several in vitro demonstrations that phosphorylation is not required for the formation of intracellular inclusions [30]. In addition to its fundamental impact upon experimental research, the answer to this question also has important implications for understanding the role of phosphorylation in synucleinopathy pathogenesis and determining if targeting kinases or phosphatases could be viable therapeutic strategies. That the two events, phosphorylation and aggregation, may influence each other, does not allow us to take one as a proxy of the other.

Following Braak’s hypothesis [8], confirmed in a way by the host-to-graft apparent pathological transmission of synucleinopathy [31,32], the α-syn prion-like hypothesis was coined. It proposes that fibrillar α-syn assemblies seed soluble α-syn into forming larger aggregates that, when they break, multiply into more seeding-potent fibrils. The seeding of α-syn pathology via the direct injection of aggregated pathological α-syn, either extracted from a patient brain or of recombinant origin, into rodent and non-human primate brains, has become a widely popular modality to experimentally address synuclein aggregative and spreading behavior as well as its resulting pathology [11,12,33,34,35,36,37]. After the pioneering work of Luk et al. [34,38], the use of preformed fibrils (PFFs) of recombinant α-syn is by far the most represented modality with hundreds of published papers. The conformational strain diversity characterizing α-syn amyloid fibrils is thought to determine the different clinical presentations of the neurodegenerative diseases underpinned by a synucleinopathy. Experimentally, various α-syn fibril polymorphs have been obtained from distinct fibrillization conditions by altering the medium constituents and were selected by amyloid monitoring using the probe, Thioflavin T.

This opinion paper is not the place for discussing the scarcity of biophysical description that accompanies PFF production in most articles (as opposed to well-controlled productions—[36,39,40]), preventing the replication of studies across laboratories, but often within the same teams when using different production batches. What deserves a discussion is a quasi-exclusive reliance upon Thioflavin T fluorescence monitoring. It is not so much about sloppiness but laziness, as most of us blindly and confidently use it without understanding how and what a rotor dye works/shows. Why do some productions “work” while others “do not”?, at least according to the now uniquely and widely adopted Thioflavin T fluorescence monitoring. Among a few examples, De Giorgi et al. recently reported that concurrently with classical Thioflavin T-positive products, fibrillization in saline also gives rise to polymorphs invisible to ThT, but visible to other rotor dyes with a trimethine cyanine skeleton [40]. The generation of Thioflavin T-visible and Thioflavin T-invisible fibril polymorphs is stochastic. It can thus skew the apparent fibrillization kinetics revealed by Thioflavin T, that binds to the outer surface of the amyloid assemblies, parallel to the main fibril axis. Interestingly, the Thioflavin T binding site differs from the binding site of dyes, such as Congo red and its X-34 derivative [41]. As polymorphs adopt different amyloid folds, the differences in the outer exposure of the C and N termini at each amyloid stack level undoubtedly affect dye binding, enabling or not Thioflavin T and the access of other trimethine cyanine skeleton dyes to their binding site (or other parameters). The emergence of these Thioflavin T-invisible polymorphs has therefore been ignored, to date, or mistaken for fibrillization inhibitions/failures. They present a yet undescribed atomic organization, and show an exacerbated propensity towards self-replication in cortical neurons. Thus, as with prion strains and luminescent conjugated polymers [42], multiple external probing using different rotor dyes could discriminate the α-syn polymorph identities, otherwise revealed by unquestionable but complex biophysical approaches, such as cryo-electron microscopy or solid-state nuclear magnetic resonance. Eventually, the injection of such Thioflavin T-invisible α-syn polymorphs into the mouse substantia nigra pars compacta, triggers a synuclein pathology consisting into true filamentous aggregates that spreads throughout the brain [40]. It is worth noting that production of Thioflavin T-invisible polymorphs has been previously communicated, but this feature was not emphasized [39], and therefore ignored.

Interestingly, reminiscent of the “ Thioflavin T dichotomy” characterizing Thioflavin T-invisible and regular α-syn polymorphs, α-syn fibrils amplified from the cerebrospinal fluid and post-mortem brain extracts of multiple system atrophy patients are, respectively ~3 and ~10 times less Thioflavin T-positive than the ones amplified from the comparable samples of PD patients [43]. In another study, α-syn fibrils amplified from brain extracts from different PD and multiple system atrophy patients also exhibited variable Thioflavin T intensities and stained differently with a set of external fluorescent probes [44]. In conclusion, we, collectively, artificially select specific polymorphs that do not reflect the diversity of the pathological polymorphs, thereby slowing down the validation of therapeutic strategies likely to fail if not designed to target those toxic species. Re-introducing a critical appraisal of our fibrillization monitoring tool is urgently needed, if we do not want to fail the next generation of clinical trials against synucleinopathies. One should keep in mind the main fundamental drawback of PFFs. Even the best characterized and most thoroughly produced PFFs are not guaranteed to induce any toxicity/pathology that is PD/multiple system atrophy-relevant in vitro or in vivo. It is likely that there is some relevance for the disease, but the way this is taken for granted in essentially all PFF-based studies sometimes should make us wonder.

Quite difficult to follow as well, is the mix up around the concepts of spreading and propagation [45,46], echoing the misconception about the validity of the Braak hypothesis. A rapid reading of the literature may suggest that α-syn is secreted [47]. This is often cited in a way that makes it sound as though α-syn is actively released in a manner similar to secreted message-carrying proteins. α-syn is a presynaptic protein involved in synaptic vesicle physiology, but does not exhibit a secretion motif and is thus not secreted *stricto sensu*. It could be passively released mostly through exosomal release [9,47,48]. The amounts at play are therefore very different, i.e., much lower in the case of non-secreted proteins, having a clear impact upon our understanding of propagation, starting with the (low) probability of prion-like templating (hence seeding) in the neighboring cells.

## 3. Conclusions and Actions

This opiniated survey does not, by far, summarize the entire list of the frustrating issues, but it gives a taste of some of the most pressing needs one feels when going through the complex literature of synucleinopathies. My research suffers from several pitfalls I have listed, and I plead guilty as charged for the accusations of sloppiness and laziness. I propose to fight against this slippery road that will undoubtedly prevent the field from moving as fast as possible.

The first action could be to avoid using the word synucleinopathy when describing the pathology in animal models of Parkinson’s disease, multiple system atrophy, or dementia with Lewy bodies. Synucleinopathy refers to a very precise (although it uses a vague terminology) anatomopathological definition corroborated by endpoints which, when used on non-human material, do not necessarily reflect the same mechanism at work. In other words, synucleinopathy should be kept for post-mortem human studies while other words should be used for experimental work. I am conscious that wording such as “synucleinopathy-related experimental model” is too long, but we shall adopt a semantic that undoubtedly differentiates patient-derived explorations from the experimental investigations.

Given the considerable literature produced every week, it becomes more critical every day to declare upfront the list of endpoints one uses in a paper: the actual endpoints and how they are utilized as proxies for something else. By such an upfront declaration, the second proposed action can be achieved in a table to be added at the end of the Materials and Methods section, immediately before presenting the statistical methods. Such a statement may allow the readers not to forget what is quantified. In addition, it would help reviewers to assess the soundness and reliability of a manuscript at the time of peer-reviewing.

Proactive lobbying of the editors of scientific journals would enable these actions to be implemented.

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
