# Peer review of "How Lazy Reading and Semantic Sloppiness May Harm Progress in Synucleinopathy Research"

_biomolecules, 2022, doi:10.3390/biom12020228_

Round 1

Reviewer 1 Report

The authors would like to show the interesting opinion on this topic.

I think that the scheme showing the overall mechanisms/systems on synucleinopathy should be presented clearly to deepen the understanding of readers.

A good figure should be added.

Reviewer 2 Report

The author has composed an entertaining and important piece on the sloppiness of how PD research is designed, reported and discussed. (The general problem is certainly not restricted to PD.) Every newcomer to the field should read this piece (and so should those that have been in the field for longer).

Strengths:

  1. The part on oversimplified and dated PD symptoms
  2. The part on pS129 as a synonym for aggregation 
  3.  The part on Braak staging (my favorite)

Weaknesses:

  1. The part on PFF models could be even more critical in my opinion. Even the best characterized and most thoroughly produced PFFs are not guaranteed to induce any toxicity/pathology that is PD-relevant in vitro or in vivo. It is likely that there is some relevance for the disease, but the way this is taken for granted in essentially all PFF-based studies sometimes makes me wonder. Also, there is little consensus if PFFs cause toxicity, e.g., in cell culture or not.
  2. The author introduces LBs as "proteinaceous, aSyn-rich inclusions". The study by Shahmoradian et al, cited by the author, questions the largely proteinaceous character of LBs, instead LBs are proposed to be "lipidous" and "organellar". "Proteinaceous" to me sounds like the classical model of Abeta plaques, consisting virtually exclusively of Abeta peptides. The author correctly points out the presence of other proteins, but the groundbreaking study by Shahmoradian et al goes beyond that by pointing at lipids/organelles and questioning the fibrillar character of aSyn in the lesions. I personally don't have a strong opinion on this matter, but if Shahmoradian et al are right (and a few others), then aSyn lesions are - at least to a large extent - not proteinaceous aggregates. While the jury is out (maybe it will never return), it seems like the proteinaceous character of LBs should be introduced with some caution.
  3. The part on PFFs is rather specific, or at least ends on a rather specific note: ThioT-invisible aggregates is sth I never heard of and, while interesting, it is hard for me to judge how big of a problem that really is. It seems though, the whole piece culminates in this problem, and the author may even have had this problem in mind when starting to write the opinion piece.
  4. The author adds that there are many other annoying issues, which he doesn't want to touch upon (lack of time or space?), but it would be nice to at least mention what they are.
  5. The remedy of avoiding the word "synucleinopathy" in animal models is definitely just a start. And I personally wouldn't be so critical about this. Synucleinopathy (pathein, Greek: to suffer), technically just means that an organism or cell suffers from synuclein. I'd be willing to accept any model in which aS causes robust harm as a "synucleinopathy" model. But that's semantics, I get that in the clinical literature a "s." seems to be well-defined. Also, this remedy won't fix the PFF problems mentioned in the text nor the sloppy clinical characterization of PD.
  6. Not all the issues mentioned may be related to sloppy reading or semantic sloppiness. The problems with PFFs, e.g., seem to be a matter of experimental design.

 I understand the author wants to focus on the issues mentioned in his piece, but I think the manuscript would benefit from at least a few critical remarks on the widely accepted assumption that aS is secreted. This is often cited in a way that makes it sound like aS is actively released like a growth factor or sth like that. With regard to PFFs, the idea of "cross-seeding" could be highlighted more.

Minor: line 141, should it be "a debate"?; line 195 "animal models of the conditions" - I don't understand this, sorry, is sth missing? Abstract and Introduction are highly similar, not sure if that's ok or not. I'll let the editor decide.

Overall, the author is to praise for this undertaking and should consider a sequel.

Round 2

Reviewer 1 Report

The authors would like not to respond the critique.

Reviewer 2 Report

I hope the next (and current) generation of PD researchers will read this piece.